# Influence of Timber Harvesting Operations and Streamside Management Zone Effectiveness on Sediment Delivery to Headwater Streams in Appalachia

**Daniel Bowker [1], Jeffrey Stringer [2] and Christopher Barton [2,*]** 

[1] Coalition for the Poudre River Watershed, 320 E. Vine Drive, Suite 121, Fort Collins, CO 80524, USA; daniel@poudrewatershed.org

[2] Department of Forestry and Natural Resources, University of Kentucky, 218 T.P. Cooper Bldg, Lexington, KY 40546, USA; jeffrey.stringer@uky.edu

\* Correspondence: barton@uky.edu

**Abstract:** Disturbances created by timber harvesting equipment and associated haul roads and skid trails can create overland sediment flows (sediment paths), especially in steeply sloping terrain, leading to stream sedimentation. This study investigated the effect of variables associated with GPS tracked harvest equipment movement, skid trail development and retirement, topography, and streamside management zone (SMZ) width and tree retention on sediment delivery to streams. While the intensity of harvest equipment traffic was not correlated with sediment path development, the presence and location of skid trails were. All of the sediment paths were found to originate at water control structures, influenced by microtopographic features, on the skid trails directly adjacent to SMZs. Mesic slopes were associated with increased sediment path development across all SMZ configurations. Two factors, the accumulation of coarse logging debris in the SMZ and the increased distance of skid trails to streams, were both correlated with decreased sediment path development. The study provides insight into how these variables interact and can be used to develop site-specific guidelines for SMZs in steeply sloping terrain that could improve their efficiency and effectiveness.

**Keywords:** sedimentation; logging; best management practices; global positioning system

## 1. Introduction

Timber harvesting has the potential to negatively impact streams and riparian habitats by increasing nutrient inputs, elevating water temperatures, and delivering sediment to streams [1,2]. The latter is the most important pollutant in many forested areas [2–4] and a significant issue for ground skidding operations. Excess sediment can have a number of negative effects on stream habitats and aquatic populations [1,2,4], while degrading drinking water and increasing the transport of sorbed pollutants, leading to higher water treatment costs [1,5].

Most sediment delivered to streams during forest operations involves haul road, skid trail, and landing construction and use [1,6,7]. In order to minimize these impacts, most states' best management practices (BMPs) include riparian or streamside management zones (SMZs), minimally disturbed buffer strips intended to filter flows of sediment and nutrients from disturbed ground and reduce the effect of canopy removal on stream temperature [8,9]. SMZs have been shown to be generally effective at reducing nutrient inputs, temperature increases, and sediment levels [1,2,10]. However, robust data on SMZ effectiveness are not available for many forest types and physiographic regions. Furthermore, the implementation of SMZs has an economic cost. The timber left unharvested in the

SMZs as well as the increased cost of navigating equipment around SMZs can reduce net revenues significantly [11,12].

One of the intentions of SMZs is to mitigate overland sediment paths, the primary conduits for sediment delivery from forest road networks to streams [13]. Sediment paths created by the concentration of sediment-laden overland flow can bypass the vegetative filtering of SMZs [13–15]. Rivenbark and Jackson [16] found that overland sediment flows generated from forest operations were capable of spanning SMZs and delivering sediments directly to streams, and were highly variable among similar sites. Fifty percent of these breakthroughs occurred in areas of convergent topography (where downhill flows of water come together), while 25% were caused by drainage from harvest roads and trails. Ward and Jackson [17] found that SMZs ameliorated sediment transport following forest harvest, reducing sediment by 71% to 99%; however, no statistical model accurately explained the variation in SMZ efficacy. A sediment routing survey by Rashin et al. [18] showed that ground disturbance within 10 m of a stream was likely to deliver sediment to the stream. White et al. [19] found that narrow SMZs effectively removed coarse textured (>20 μm) sediments from concentrated overland flows of runoff water, while a wider 16 m SMZ removed the majority of 2 to 20 μm sediments. Litschert and MacDonald [20] found that the length of sediment delivery pathways in the Sierra Nevada and Cascade Mountains of California was significantly related to mean annual precipitation, the cosine of the aspect, elevation, and hillslope gradient. Eighty-three percent of these pathways connected to the stream network originated from a skid trail [20]. A study in the Virginia Piedmont, testing the efficacy of three different SMZ buffer widths, found that wider SMZ buffers were no more effective than narrower buffers in preventing overland sediment flows from reaching streams, and the sediment pathways found passed through the SMZ regardless of width [15]. In all cases, these flows were caused by failed water control structures on steep slopes with fragile soils.

Many states' SMZ recommendations stem from research carried out in the White Mountains of New Hampshire in the 1950's, with regional variations based upon differences in geology, soils, and harvesting systems [6]. However, little research has been completed to tailor these recommendations to specific regions or sites, and only a few studies have looked into the efficacy of different buffer widths and canopy retention levels [9,13]. In fact, the lack of region and site-specific information about SMZ buffers is partly responsible for the wide variations in SMZ regulations among states [8]. Most research that has been conducted on BMPs and SMZs has been directed at larger order streams [18,21]. Regulatory efforts have largely ignored lower ordered streams [22], even though they can comprise 60–80% of the drainage network and are major contributors of sediment and wood to larger streams [21,23]. There is a need for research into the mechanisms delivering sediment to lower ordered streams [24,25], and specifically how forest harvesting machinery and its associated transportation network deliver this sediment [3].

This study attempts to determine the effect of SMZ elements, harvesting equipment activity, skid trails associated with ground skidding operations, logging debris, and topography on indicators of sediment delivery in the highly dissected region of the upper Cumberland Plateau in Appalachia. This study uniquely employed the use of GPS tracking of all harvesting equipment to assist in accurately describing the impacts of equipment movement on sediment delivery to streams. In particular, variable SMZ widths and tree retentions were evaluated for their effectiveness at reducing sediment breakthrough along intermittent and perennial headwater streams. Along with individual equipment movement patterns, ground disturbances, including the skid trail network, and topographic parameters were also examined for their propensity to facilitate or inhibit sediment paths.

## 2. Materials and Methods

The study took place on the University of Kentucky's Robinson Forest, a 6070 ha experimental forest located in Breathitt, Knott, and Perry Counties in eastern Kentucky. Robinson Forest is in the rugged interior section of the Cumberland Plateau (longitude 83.14° W, latitude 37.47° N), highly dissected, steeply sloped (generally 40% to 80%) ranging in elevation from 244 to 488 m above mean

sea level [26]. The geology consists of interbedded sandstone, siltstone, shale, and coal, while the three major soil complexes in the study area are classified as severely erodible, and as poorly suited for roads [27]. It is predominantly occupied by 100-year-old even-aged mixed mesophytic and oak-hickory forest types [27,28]. The 26-year annual precipitation average at Robinson Forest is 117.6 cm with an average monthly total of 9.9 cm, March being the wettest on average and October the driest.

This study is a part of a larger research collaboration investigating the effects of SMZ configurations of width and overstory retention on water quantity and quality on intermittent and perennial streams [29–31]. Operationally, this SMZ project encompassed commercial timber harvests of 6 forested watersheds with 2 replicates of 3 different SMZ configurations (Table 1), with 2 unharvested controls. Of the 6 treated watersheds, the 3 Shelly Rock Fork watersheds of 27, 72, and 33 ha in size along with two control watersheds of 97 and 79 ha were used in this study. These three harvested watersheds provided six variants of SMZ width and percent dominant/codominant overstory retention as indicated in Table 1. For intermittent streams, these included 7.6 m width and 0% overstory retention (7.6 m/0%), 7.6 m/25%, 15.2 m/25%, and for perennial streams 16.8 m/50%, 16.8 m/100%, and 33.5 m/100% (see [31] for additional information on study design).

**Table 1.** Streamside management zone (SMZ) treatment variants by stream type, including SMZ width, percent canopy retention and the number of fixed area experimental units established to correlate sediment path occurrence with equipment operation intensity and sediment path source.

| Treatment | SMZ Width (m) | Canopy Retention (%) | Experimental Units (Number) |
|---|---|---|---|
| I1 | 7.6 | 0 | 3 |
| I2 | 7.6 | 25 | 4 |
| I3 | 15.2 | 25 | 10 |
| P1 | 16.8 | 50 | 3 |
| P2 | 16.8 | 100 | 4 |
| P3 | 33.5 | 100 | 4 |

### 2.1. Harvesting

Timber harvesting was conducted in these three watersheds from June 2008 to March 2009, and was performed by one logging firm using a Timbco 445EXL track mounted, swing-arm feller buncher; John Deere 650, 700, and 850 bulldozers; and Caterpillar 525 cable and 545 grapple log skidders. All timber was skidded uphill to log decks located on the tops of ridges and hauling was conducted outside of the study watersheds. Constructed primary skid trails were installed from the landings primarily along the contour at intervals from the edge or near the edge of the SMZ boundary upslope to the ridge. All merchantable material was removed with the exception of 2.3 to 3.4 $m^2\,ha^{-1}$ of dominant and codominant reserve trees outside of the SMZ areas [32]. Retention within the SMZ was marked and harvested according to Table 1. Harvesting involved chainsaw felling of the majority of timber within the SMZ, limited skidding within the SMZ and no skid trail construction. Merchantable material was cabled from the SMZ where needed using a bulldozer or cable skidder and none of the intermittent or perennial streams were crossed, maintaining the integrity of the SMZ treatments. The vast majority of the ground skidding was accomplished on primary or secondary constructed skid trails with only limited use of non-constructed tertiary trails. Completion of the harvest involved the retirement of all skid trails, landings, and haul roads, including removing ruts, installing permanent water control structures on constructed skid trails, and other associated retirement work, such as seeding all disturbed ground with an orchard grass and winter wheat seed mix [33].

### 2.2. Equipment and Sediment Path Monitoring

To facilitate the correlation of the intensity of harvest equipment use on sediment delivery, MultiDAT Jr. GPS receivers (Castonguay Electronique, Longueuil, QC, Canada) were installed on

the feller buncher, bulldozers, and wheeled skidders. GPS-equipped MultiDATs were set to take a position every 30 s while the machines were operating using machine specific maximum vibration thresholds to trigger data collection. MultiDAT data were retrieved weekly using an iPAQ Pocket PC (Hewlett-Packard Company, Palo Alto, CA, USA) and downloaded into MultiDAT version 5.1.3 software, then exported in ArcGIS shapefile format for analysis with versions 9.2 and 10 of ArcGIS Desktop (ESRI, Redlands, CA, USA). Other shapefiles necessary for analysis (for example, skid trail and water control structure locations) were produced using a Trimble GeoXM handheld GPS unit with the GPScorrect differential correction extension (Trimble Navigation Limited, Sunnyvale, CA, USA) running ArcPad 7.0 (ESRI, Redlands, CA, USA).

Within one year after completion of the harvest, perennial and intermittent stream channels within the three harvested and two unharvested watersheds were scouted for the presence of overland sediment flow not associated with established ephemeral channels. The steep and dissected topography resulted in all sediment flows being concentrated as distinct sediment paths, and only paths that contacted the stream channel were assessed. To facilitate the analysis of variables associated with sediment path occurrence, experimental units were established (Table 1) as roughly rectangular plots bordering stream segments that encompassed an area with similar topography, harvest, and SMZ characteristics, and created as polygon shapefiles in ArcMap (Figure 1). Each unit encompassed a section of perennial or intermittent stream along with slopes and skid trails above the stream. All units were drawn to encompass at least one section of primary skid trail and all variables were assessed within these experimental units.

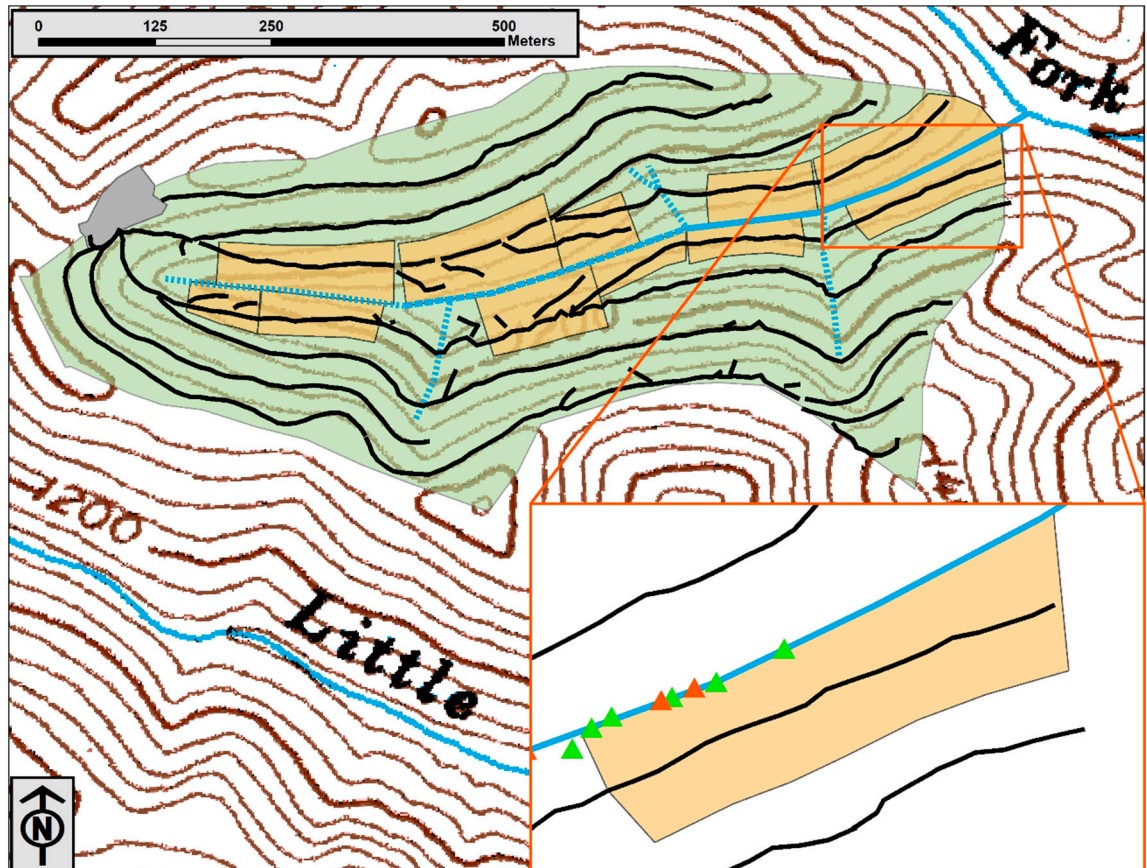

**Figure 1.** Map of the treatment 2 watershed (green) indicating experimental units (tan), perennial (solid blue) and intermittent streams (bold dashed blue), ephemeral channels (hatched blue), constructed skid trails (black lines) and log landing (grey). Inset of one of the experimental units shows triangles denoting the location of sediment paths entering the stream (green from north facing slope, red from south facing slope.).

Each of the sediment paths was GPS located and the following information recorded: width of the path contacting the stream; source type (primary, secondary, or tertiary skid trail, off trail equipment operation indicated by forest floor disturbance, areas with no indications of forest floor disturbance); skid trail microtopography at the source (e.g., sloping [>5%], non-sloping [<5%], low point [positive slope in both directions]; water control structure type at source [waterbar, dip, trench, berm cut]); and slope distance and degree from the source to the stream. It is important to note that pre-harvest ephemeral channels existed in the study area. Skid trails crossed these channels and were documented sources of sediment as reported by Bowker [29] and Witt et al. [30]. However, these are not considered overland sediment flow paths and were not included in this study. Only sediment paths that were directly associated with harvest operations and equipment activity are included in the study. All of the variables were entered into an ArcGIS shapefile attribute table and exported to a Microsoft Excel spreadsheet (Microsoft Corporation, Redmond, WA, USA). Statistical analysis was performed using JMP versions 9 and 11 (SAS Institute Inc., Cary, NC, USA). Due to the unequal sample sizes, means comparison was carried out by the Tukey–Kramer honest significant difference test. Linear models and linear regressions were used in order to determine significant factors associated with sediment path initiation, and were created by the standard least squares procedure, with pairwise multivariate analyses run to identify and eliminate highly correlated variables. Significance was set at the $\alpha = 0.05$ level.

An attribute table for the experimental units was created in ArcMap to characterize each unit as to the variables that were associated with, and that may influence, sediment path initiation. SMZ buffer width and canopy retention percent were entered, as well as treatment designation number 0 (control), 1, 2, or 3. ArcMap was used to determine experimental unit area; stream length; length of primary, secondary and tertiary skid trails; total skid trail length; and the minimum skid trail to stream distance. Mean and maximum slope degree and aspect for each unit were derived from 10 m digital elevation model (DEM) data, as was the average aspect. Aspect was transformed to a moisture index from 0 to 2 (0 = xeric/subxeric southwest slopes, 2 = mesic/submesic northeast slopes), with 1 representing an intermediate northwest or southeast facing slope [34]. The total number of harvest machine traffic GPS positions within the experimental unit was entered into the attribute table as a total traffic intensity value for that unit. An index value of machine traffic intensity was determined by dividing this total traffic intensity value by the area of the unit. A skid trail density index value was determined by multiplying the total length of skid trail within each unit by an average skid trail width (4.9 m), then dividing the total area of skid trail within the analysis unit by the area of the unit itself. An SMZ and upslope post-harvest residual basal area for each experimental unit was determined by point sampling with a 10 factor prism, to provide an average for the experimental unit.

The response variable for the analysis of treatment effects and linear modeling was the number of sediment paths directly related to equipment operations within each experimental unit and expressed per 305 m of stream. The total width of sediment paths within each experimental unit was also calculated.

### 2.3. Logging Debris

The characterization of logging debris contacting the forest floor within each experimental unit was determined. Logging debris was the result of delimbing and topping operations within the SMZ that occurred at the stump. Logging debris was not accumulated or manipulated. Logging debris was characterized using randomly located transects within experimental units. Transects ran from the skid trail to the stream and at 3 m intervals logging debris (tops and branches) contacting the forest floor was classified as fine (<10.2 cm diameter) or coarse (>10.2 cm diameter) [35] and assigned a value of 1 (presence) or 0 (absence). A debris index was determined for each transect indicating occurrence by size class and averaged to yield a mean debris index value for each experimental unit.

## 3. Results

### 3.1. GPS Performance

A total of 680,227 GPS locations were recorded during the course of the harvest of the three watersheds: 272,303 bulldozer, 127,821 feller buncher, and 280,392 skidder positions were recorded. GPS locations obtained from the MultiDAT units lined up well with the GeoXM-captured skid trails. A raster map was created in ArcGIS to show the relative traffic intensity for all harvest machines along the skid trail network (Figure 2). The MultiDAT GPS dataloggers were found to be an effective tool to obtain the overall pattern of machine movement, along with the intensity of that movement along certain sections of skid trail.

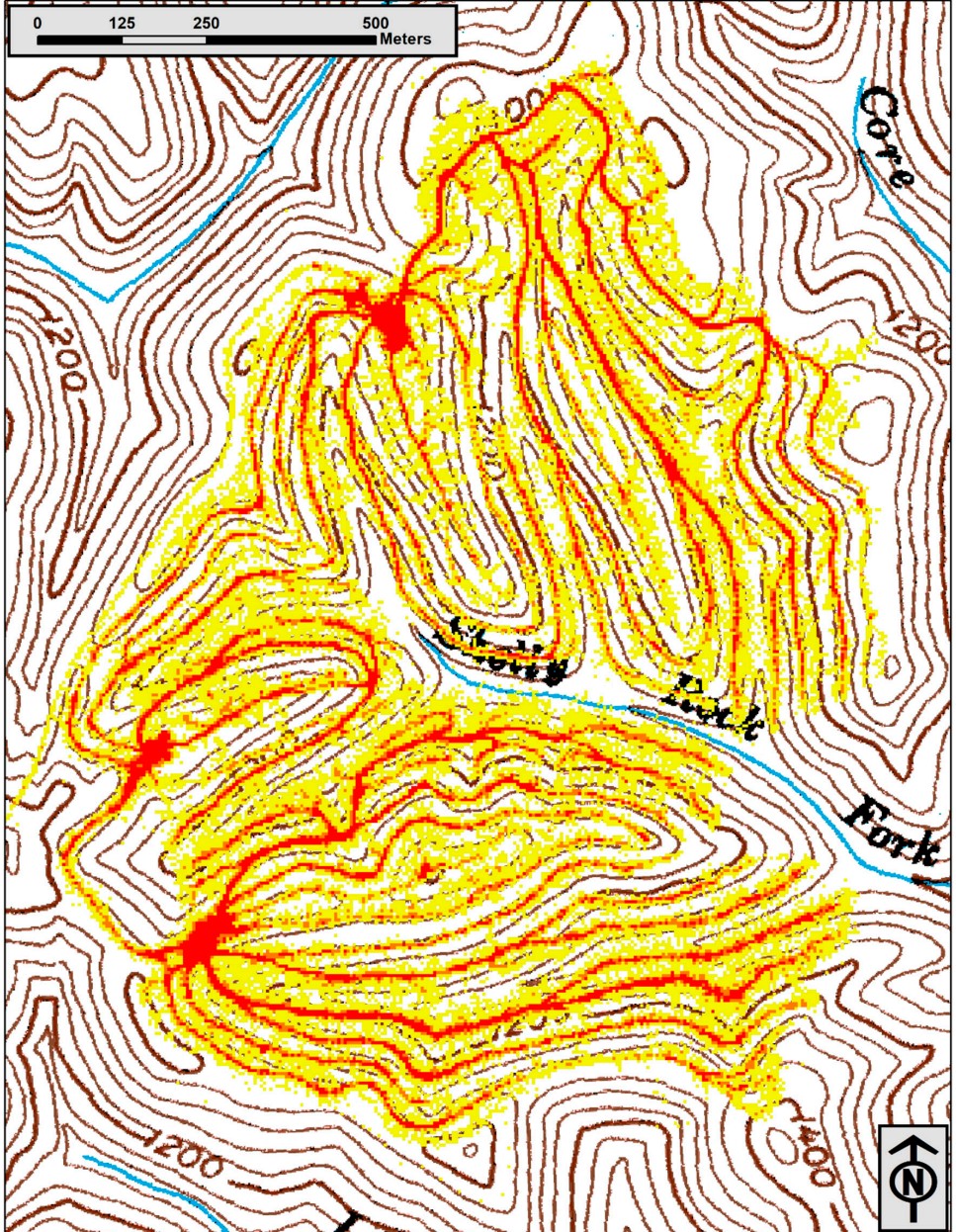

**Figure 2.** Raster map showing relative traffic intensity for all harvest machines on the three study watersheds. Yellow cells are least trafficked, orange are more trafficked, and red are the most trafficked skid trail sections.

### 3.2. Sediment Paths

There was a total of 72 sediment paths found within the experimental units, which all originated from a skid trail. Thirty-five of the 72 paths originated from primary skid trails, 35 from secondary trails, and only two from tertiary skid trails. This pattern is to be expected, as the tertiary skid trails were non-bladed and generally were not observed to have as much exposed soil surface as the constructed skid trails. No sediment paths were found to originate from equipment disturbance off skid trails. Sediment paths ranged 2.4 to 57.6 m in length (mean = 20.6 m) and 0.2 to 10 m in width (mean = 2.3 m), with slopes ranging 25 to 48 degrees and averaging 34.1 degrees, reflective of the steep terrain associated with this physiographic region. The widest paths appeared in very steep terrain near the edge of the floodplain and were the result of mass wasting, likely caused by water movement in the sediment paths.

### 3.3. Water Control Structures

All of the 72 sediment paths were associated with a water control structure installed during skid trail retirement. Reverse grade structures, including waterbars and broad-based dips, were the most commonly used water control structures. Berm cuts were less commonly used, though four were associated with a sediment path. A total of 479 reverse grade structures were found within the experimental units with 68 of these (14.2%) associated with a sediment path reaching the stream (Table 2). Of the 479 total structures, 155 were placed on relatively flat skid trail sections (<5% slope), with only 8.3% (13) producing a sediment path. There were 312 structures constructed along sloping skid trails (>5% slope), with 14.4% (45) producing a sediment path that reached the stream. Only 12 of the total 479 reverse grade structures (in this case waterbars) were constructed at the microtopographic low contour point of a skid trail with positive skid trial slope in both directions from the waterbar. However, these 12 were of interest, because 83.3% (10) were associated with a sediment path. While none of these reverse grade structures were poorly constructed in themselves, locating them at a low point allowed for water to concentrate along the trail slope from both directions, resulting in a sediment path.

**Table 2.** Water control structures within experimental units of the three harvested watersheds, by the morphology of the skid trail where they were constructed. Percentage of water control structures associated with initiation of machine-caused sediment paths is also shown.

| Skid Trail Morphology | Number of Structures | Number that Produced a Sediment Path (% of Total) |
| :---: | :---: | :---: |
| Flat (<5% slope) | 155 | 13 (8.3) |
| Low point | 12 | 10 (83.3) |
| Sloping (>5% slope) | 312 | 45 (14.4) |
| Total | 479 | 68 (14.2) |

### 3.4. Treatment Effects

The analysis of sediment path number and width by treatment yielded similar statistical significance; thus, only the results of sediment path number analysis are presented. Analyzing the intermittent stream sections as a group, there was a trend—but no statistical difference—in sediment path number between SMZ widths of 7.6 m, regardless of having 25% (I2) or 0% (I1) overstory retention (10.0 and 3.6 paths per 305 m, respectively) (Table 1). However, when the intermittent SMZ width increased to 15.2 m with 25% overstory retention (IT3), only 1.5 paths per 305 m were observed which was significantly lower ($p = 0.0275$) than the 7.6 m/0% (I1) treatment, indicating a positive effect associated with increasing SMZ width and retention.

Compared to the 16.8 m/50% perennial stream P1 treatment, the 16.8 m/100% P2 treatment produced a significantly greater ($p = 0.0014$) number of sediment paths (0 and 6.9 per 305 m, respectively) (Table 1). This was associated with a significant decrease ($p = 0.0199$) in coarse logging debris index, 13 for the

16.8 m/100% P2 treatment as compared to 83 for the 16.8 m/50% P1 treatment (Table 3). No significance was found for fine or total logging debris indices. However, the negative relationship between coarse logging debris (>10.2 cm diameter) and sediment path number indicates a potential causal effect of the debris inhibiting sediment paths reaching the stream for SMZ widths of 16.8 m. In contrast the 33.5 m width P3 treatment that retained 100% overstory had statistically fewer ($p = 0.0012$) sediment paths (0.4 per 305 m) than P2 and was similar to P1. However, the P3 treatment had a low logging debris index (25), similar to the P2 overstory retention treatment. As no trees were felled in the 100% retention perennial SMZs, any logging debris within the SMZ was derived from tops originating in upslope harvested areas. This suggests that the increased SMZ distance from 16.8 m to 33.5 m was responsible for mitigating sediment path connectivity with the stream without a significant presence of logging debris.

**Table 3.** Mean number of sediment paths and coarse logging debris index by stream type and SMZ treatment variants. Values with the same letter are not significantly different ($p < 0.05$) within each stream type.

| Treatment | Skid Trail Distance to Stream Bank (m) | Mean Number of Sediment Paths | Coarse Debris Index |
|---|---|---|---|
| I1 | 7.6 | 10.0a | 70a |
| I2 | 7.6 | 3.6ab | 63a |
| I3 | 15.2 | 1.5b | 40a |
| P1 | 16.8 | 0.0b | 83a |
| P2 | 16.8 | 6.9a | 13b |
| P3 | 33.5 | 0.4b | 25b |

Linear models were created to investigate the importance of the effects the independent variables had on the number of sediment paths per 305 m of stream. Independent variables included: treatment prescribed SMZ width, average slope of the analysis unit, moisture index, residual basal area, logging debris index, traffic intensity, trail density, and minimum distance from skid trail to stream in the experimental unit. Pooling the modeling for perennial and intermittent stream sections explained approximately 46% of the variability in number of sediment paths ($R^2 = 0.46$, overall model probability of $p < 0.01$). Significant variables included moisture index ($p = 0.03$) with mesic sites positively associated with sediment path number; greater distance between trail and stream ($p = 0.01$) resulted in fewer paths; while coarse logging debris ($p = 0.04$) and sediment path number displayed a negative relationship. In all model variations, traffic intensity within the analysis unit was not an important factor in number of sediment paths. This included trafficking of bulldozers, skidders, and the feller buncher on and off the skid trails in the analysis units.

## 4. Discussion

The results of this study are useful in understanding the dynamics and spatial distribution of overland sediment delivery mechanisms associated with timber harvesting in steep terrain. In particular, all of the machine-caused sediment paths observed in this study originated from skid trails. There was no relationship between sediment path occurrence and off-trail equipment use, or the intensity of equipment use on skid trails, nor did sediment paths originate from the general harvest area. In a study by Rivenbark and Jackson [16], only 25% of the SMZ breakthroughs they observed were caused by drainage from the trail network, whereas Litschert and MacDonald [20] found that five of six pathways (83%) originated from the trail network. This study found that 100% of sediment paths were associated with water control structures, which is in close agreement to Litschert and MacDonald [20]. In fact, these structures may have actually concentrated the flow and increased the likelihood of sediment reaching the stream network, as also described by Lakel et al. [15]. Our results found specific contour or microtopographic features that were prone to producing sediment paths capable of reaching streams,

and low points along skid trails where water accumulated from two directions exacerbated sediment path development. While the mitigation of sediment path development can be accomplished in skid trail construction by reducing the number of accumulation points, in highly dissected terrain there may be instances where this is unavoidable. In these cases, the proper placement of multiple water control structures to collect runoff before it accumulates would likely reduce sediment path development.

The issue of buffer width and canopy retention to reduce overland sediment delivery was a major focus of our study. In the Piedmont physiographic of Virginia, Lakel et al. [15] found that wider (30.4-m) SMZ buffers were no more effective in preventing overland sediment delivery than narrower (15.2-m) buffers. On the other hand, Swift [7] found that wider (50-m) SMZ buffers were effective at preventing a greater proportion of sediment delivery to streams than narrower (10-m) buffers in the steeply sloped Appalachian Mountains in North Carolina. This study corroborates Swift's findings, providing evidence that increased buffer width helped prevent sediment paths in the intermittent and perennial stream reaches. Interestingly, fewer sediment paths were observed in perennial P1 treatments over that observed in P2, where 100% of the canopy was retained. In a study that evaluated the effectiveness of the SMZs for protecting water quality [31], P1 storm total suspended solid (TSS) levels were much higher than those for the other treatments examined. Storm TSS concentrations and turbidity readings in P2 and P3 watersheds were statistically similar to those in unharvested control watersheds. Because the BMP prescriptions varied by stream type within a watershed, it was difficult to discern whether an individual recommendation was responsible for the observed reduction or whether it was the result of combined practices. For example, it was noted in a separate study examining ephemeral stream crossings in these watersheds that TSS concentrations from two types of improved crossings (skidder bridges and culverts) used in P2 and P3 watersheds were statistically similar to concentrations in unharvested control watersheds, whereas P1 watersheds that used earthen fords were a significant contributor of storm TSS [30]. The fewer sediment paths observed in this study in the P1 perennial treatment over that of P2 further suggest that the unimproved crossings (fords) in ephemeral channels are the major contributor to the observed elevated TSS concentrations.

The present study also found an interesting and operationally useful interactive relationship between skid trail distance to stream and the occurrence of coarse logging debris. The study showed that normally distributed logging debris from delimbing and topping operations was correlated with a significant reduction in sediment path occurrence when skid trails occurred 16.8 m from streams (P1 and P2). Litschert and MacDonald [20] also reported that logging debris had a negative effect on the ability of an overland sediment pathway to reach the stream network. However, in our study, when skid trail distance to the stream was reduced to 7.6 m, as was the case of intermittent stream widths in I1 and I2, there was no reduction in sediment path occurrence with increased coarse logging debris supporting. Rashin et al. [16] that found ground disturbance within 10 m of the stream was likely to result in sediment transport to the stream. In our case, scattered logging debris from cutting 75% of the overstory could not mitigate sediment delivery. In close proximity, as was the case in P1 and P2, constructed debris or brush barriers proposed by Swift [7] may be required to reduce sediment transport to streams. In contrast, when the distance to stream increased in our study to 33.5 m in P3, there was no effect of logging debris on sediment path development, indicating that distance alone could reduce sediment delivery without the presence, or need, of logging debris. These findings indicate that normally distributed logging debris can be used, or can be expected, to mitigate sediment deposition in streams with 16.8 m SMZ widths. As skid trail distance decreases to 7.6 m, scattered logging debris cannot be expected to reduce sediment delivery. Likewise, when skid trails are placed 33.5 m from the stream, logging debris is not required to mitigate sediment delivery, as distance is sufficient. These results have interesting implications and provide useful information for the development of requirements and guidelines for BMPs in general, and SMZs specifically, for timber harvesting operations in steep terrain.

Moisture index, a function of the aspect of a site, showed a significant positive relationship in this study to number of machine-caused sediment paths. A similar finding is reported by Litschert and

MacDonald [20], where the length of overland flow pathways was significantly related to the cosine of the aspect of the hillside.

## 5. Conclusions

Overall, this study shows that site-specific SMZ guidelines can improve SMZ performance. Our study indicated that the relationship between slope aspect and soil moisture, buffer width, placement of skid trail water control structures, and logging debris accumulation affect the generation of sediment paths, demonstrating the importance of site specificity in BMP guideline development. We found that trail building on mesic slopes led to more sediment reaching streams, suggesting the value of increasing water control structure frequency on constructed skid trails, especially adjacent to SMZs on mesic slopes. Buffer strip width, the level of harvesting within the buffer and the resulting coarse logging debris in contact with the forest floor are all interrelated and associated with sediment path development. Our results suggest that a 33.5 m buffer in steep terrain, where no harvesting occurs and in the absence of logging debris, may be able to provide adequate control of sediment paths. Alternatively, a narrower buffer may work well if sufficient logging debris is left between the trail and stream, skid trail construction minimizes runoff accumulation points, and water control structures are sited and constructed well. The latter was found to be more important than the intensity of equipment use along and in proximity to skid trails. Importantly, Witt et al. [30] found that unimproved skid trail crossings of ephemeral channels in these same watersheds resulted in increased sediment delivery to streams, and that careful consideration of crossing practices must also be employed to minimize total sediment delivery from harvesting activities. Our results provide information that can be used to assess SMZ criteria and, in some cases, provide alternative approaches that are site and operation specific to improve the efficacy of BMPs during timber harvesting operations.

**Author Contributions:** Conceptualization, J.S. and C.B.; methodology, J.S., D.B. and C.B.; formal analysis, D.B.; writing—original draft preparation, D.B.; writing—review and editing, D.B., J.S. and C.D.; funding acquisition, C.B and J.S. All authors have read and agreed to the published version of the manuscript.

**Funding:** Funding for this project was provided by the University of Kentucky College of Agriculture, Food and Environment's SB 271 Water Quality Program. The work was also supported by the National Institute of Food and Agriculture, U.S. Department of Agriculture, McIntire-Stennis Research Program (Accession Number 1005547).

**Acknowledgments:** We express our gratitude to the staff at Robinson Forest for helping with many aspects of this study.

**Conflicts of Interest:** The authors declare no conflict of interest.

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
