# Peer review of "Influence of Timber Harvesting Operations and Streamside Management Zone Effectiveness on Sediment Delivery to Headwater Streams in Appalachia"

_forests, doi:10.3390/f11060623_

Round 1

Reviewer 1 Report

Overall: This manuscript is well-written and provides valuable information about the relationship between timber harvesting, best management practices implementation, and water quality. I suggest a few changes to clarify the comparisons being made between this study and previous ones.

Line 18: What about skid trails with sediment path development? Skid trail presence? Skid trail location?

Line 34: Define abbreviations. Most people know what BMP stands for, but it is still best practice to define abbreviations.

Line 39: Comma after “further”.

Lines 51-64: Consider highlighting changes in BMP recommendations over time and the time frame of the studies.

Line 65: SMZs are non-regulatory in many states.

Line 91: A range of slopes might help readers from other readers contextualize “steeply sloped”.

Table 1: It might be helpful to have different titles from intermittent and perennial stream treatments.

Figure 1: Scale is in feet whereas the rest of the manuscript uses SI units. Otherwise, a nice figure.

Line 218: The semicolon should be a colon. Colons precede lists; semicolons separate two independent clauses.

Figure 2: Again, good figure except that the scale is in feet instead of SI units.

Line 239: It may be helpful to readers to define “modified” broad-based dips.

Table 2: Consider adding parenthetical definition of “flat” and “sloping” so that readers can determine what qualified for these categories without consulting the text.

Lines 243-244: Do not begin sentences with Arabic numerals.

Table 3: “Letters indicate statistically significant differences within stream type” should be reworded. For example, “Within each stream type, values in columns not connected by the same letter are statistically significantly different at alpha = 0.05.”

Lines 301-303: Were the trails observed by Rivenbark and Jackson and Lischert and MacDonald bladed skid trails or overland skid trails? This is an important distinction.

Lines 312-318: It would be helpful to identify the widths investigated in these studies as well as the topography in the study areas. Going from a 50’ buffer to a 100’ buffer might not matter in the Piedmont or Coastal Plain but might make a big difference on the Cumberland Plateau. Likewise, the presence of bladed skid trails might necessitate wider buffers that would be unnecessary with overland skid trails.

Line 400: What year was this published?

Author Response

Response to reviewer’s comments on manuscript:

Influence of Timber Harvesting Operations and Streamside Management Zone Effectiveness on Sediment Delivery to Headwater Streams in Appalachia

Reviewer 1

Line 18: What about skid trails with sediment path development? Skid trail presence? Skid trail location?

Response: Both the presence and location of skid trails influenced sediment path presence. Sentence changed to (lines 17-18): While the intensity of harvest equipment traffic was not correlated with sediment path development, the presence and location of skid trails were.

Line 34: Define abbreviations. Most people know what BMP stands for, but it is still best practice to define abbreviations.

Response: changed to “best management practices (BMPs)”

Line 39: Comma after “further”.

Response: Changed as requested.

Lines 51-64: Consider highlighting changes in BMP recommendations over time and the time frame of the studies.

Line 65: SMZs are non-regulatory in many states.

Response: That is correct. Changed “regulated” to “recommended”.

Line 91: A range of slopes might help readers from other readers contextualize “steeply sloped”.

Response: Included the following range (generally 40 to 80%). On line 92.  Later in the paper we describe specific slopes for the sediment paths (line 236): “slopes ranging 25 to 48 degrees and averaging 34.1 degrees”.

Table 1: It might be helpful to have different titles from intermittent and perennial stream treatments.

Response: Intermittent treatments changed from T1, T2 and T3 to I1, I2 and I3; Perennial treatments changed from T1, T2 and T3 to P1, P2 and P3. Changes were made to reflect the change throughout text.

Figure 1: Scale is in feet whereas the rest of the manuscript uses SI units. Otherwise, a nice figure.

Response: Scale on map was changed to SI units.

Line 218: The semicolon should be a colon. Colons precede lists; semicolons separate two independent clauses.

Response: Changed as requested.

Figure 2: Again, good figure except that the scale is in feet instead of SI units.

Response: Scale on map was changed to SI units.

Line 239: It may be helpful to readers to define “modified” broad-based dips.

Response: Modified refers to them being adjusted in length based upon slope and road conditions, which is a common practice. As such, we deleted the word “modified” as we recognize that variation in the design is common.

Table 2: Consider adding parenthetical definition of “flat” and “sloping” so that readers can determine what qualified for these categories without consulting the text.

Response: Added (< 5% slope) after Flat and (> 5% slope) after Sloping in the table.

 Lines 243-244: Do not begin sentences with Arabic numerals.

Response: Changed as suggested (lines 244-247). “Of the 479 total structures, 155 were placed….” and “There were 312 structures…..”.

 Table 3: “Letters indicate statistically significant differences within stream type” should be reworded. For example, “Within each stream type, values in columns not connected by the same letter are statistically significantly different at alpha = 0.05.”

Response: Values with the same letter are not significantly different (p<0.05) within each stream type.

 Lines 301-303: Were the trails observed by Rivenbark and Jackson and Lischert and MacDonald bladed skid trails or overland skid trails? This is an important distinction.

Response: Neither study explicitly states how the skid trails were created or retired (or if bladed or overland skid trails). Both studies were conducted at least a year after the harvesting. In the case of the Lischert and MacDonald study, some of the study locations were many years after harvest and they note that surveys similar to those they conducted should be done within 5 years of harvest (Rivenbark and Jackson recommend within 2 years of harvest)

Lines 312-318: It would be helpful to identify the widths investigated in these studies as well as the topography in the study areas. Going from a 50’ buffer to a 100’ buffer might not matter in the Piedmont or Coastal Plain but might make a big difference on the Cumberland Plateau. Likewise, the presence of bladed skid trails might necessitate wider buffers that would be unnecessary with overland skid trails.

Response: Added information about study locations and widths compared in lines 327-331: “In the Piedmont physiographic of Virginia, Lakel et al. [15] found that wider (30.4-m) SMZ buffers were no more effective in preventing overland sediment delivery than narrower (15.2-m) buffers. Whereas, Swift [7] found that wider (50-m) SMZ buffers were effective at preventing a greater proportion of sediment delivery to streams than narrower (10-m) buffers in the steeply sloped Appalachian Mountains in North Carolina.”

Line 400: What year was this published?

Response: 1957…corrected.

Reviewer 2 Report

Review of the manuscript “Influence of Timber Harvesting Operations and Streamside Management Zone Effectiveness on Sediment Delivery to Headwater Streams in Appalachia”.

General comments:

Negative impact of intensive forest harvesting including timber transport to level of erosion and sediment path is a serious issue in forest management and utilization of natural resources. Therefore looking for methods how to mitigate such negative impacts seems to be very important.

However, this study is unbalanced to a certain extent and the chapters Results and Discussion are inconsistent. Moreover, a part of results is missing. See specific comments.

Specific comments:

Page 1, Line 34. First use of the abbreviation BMP (best management practice) must be explained. The study is for more readers than only for American forest experts who know such abbreviation.

Page 9, Line 234. Authors mentioned that width of sediments paths had varied from 0,2 m to 10 meters. The sediment path wide 10 meters is a disaster heavily influencing water quality in a stream. So I would like to see a picture of such wide sediment path.

Chapter results. Authors introduce information that they conducted evaluation of relationship among lot of variables. However, in the study is no table od correlation coefficients. Such tables must be displayed. Authors also mentioned linear regression but in the chapter there are no figures of such regressions.

Page 13, Line 328. What are C watersheds. I do not understand the abbreviation.

Page 14, Line 355. Relationship of the moisture index to number of sediment path is introduced in this chapter but results (table) are missing in the chapter Results.

Chapter Discussion. Authors broadly discussed issue of total suspended solids (TSS) (Lines 320-331) but their study did not deal with this aspect. So such part needn’t be so broad or the connection of TSS with this study must be explained.

Chapter Discussion. Authors shows differences among skid trail morphology and number of sediments path (Table 2). Flat parts of trails cause the least number of sediment path. This statement is not discussed. Discussion is concentrated to width of streamside management zone or presence of logging debris but an easy way how to mitigate negative impact of timber transport is neglected. More careful design of forest skid trails whose slope gradient is less than 5% (flat) enables relatively harmless timber transport from clear-cut areas to log landings. Authors ignored such opportunity although they mentioned the fact that primary skid trails were installed along the contour (Page 4, Line 112).

Chapter Conclusion: Authors introduces that they found that harvesting and trail building on mesic slopes will lead to more sediment reaching streams (Page 14, Lines 364-365). However, this statement is not supported in their study. Only trail building lead to more sediment reaching streams. Harvesting means operations from tree felling to timber transport. But authors say that no sediment paths were found to originate from equipment disturbances off skid trails (Page 9, Lines 233-234).

Author Response

Response to reviewer’s comments on manuscript:

Influence of Timber Harvesting Operations and Streamside Management Zone Effectiveness on Sediment Delivery to Headwater Streams in Appalachia

Reviewer 2    

Page 1, Line 34. First use of the abbreviation BMP (best management practice) must be explained. The study is for more readers than only for American forest experts who know such abbreviation.

Response: Corrected as suggested

Page 9, Line 234. Authors mentioned that width of sediments paths had varied from 0,2 m to 10 meters. The sediment path wide 10 meters is a disaster heavily influencing water quality in a stream. So I would like to see a picture of such wide sediment path.

Response: Although not common, we did observe a few sediment paths that resulted in a slump (or mass wasting) at the edge of the floodplain. These were always associated with very steep topography and the water movement in the sediment path likely triggered the development. We made a note on lines 237 – 239: “The widest paths appeared in very steep terrain near the edge of the floodplain and were the result of mass wasting, likely caused by water movement in the sediment paths.”  

Chapter results. Authors introduce information that they conducted evaluation of relationship among lot of variables. However, in the study is no table od correlation coefficients. Such tables must be displayed. Authors also mentioned linear regression but in the chapter there are no figures of such regressions.

Response: Throughout the Results section p-values for statistical significance and R2 values for regression correlation are presented (see Section 3.4). Table 3 also contains information on statistical significance.

Page 13, Line 328. What are C watersheds. I do not understand the abbreviation.

Response: C refers to unharvested control watersheds. We corrected in text.

Page 14, Line 355. Relationship of the moisture index to number of sediment path is introduced in this chapter but results (table) are missing in the chapter Results.

Response: See lines 291 – 299 where it states: “Independent variables included: treatment prescribed SMZ width, average slope of the analysis unit, moisture index, residual basal area, logging debris index, traffic intensity, trail density, and minimum distance from skid trail to stream in the experimental unit. Modeling for perennial and intermittent stream sections pooled together resulted in explaining approximately 46% of the variability in number of sediment paths (R2 = 0.46, overall model probability of p < 0.01). Significant variables included moisture index (p=0.03) with mesic sites positively associated with sediment path number; greater distance between trail and stream (p=0.01) resulted in fewer paths; while coarse logging debris (p=0.04) and sediment path number displayed a negative relationship.”

Chapter Discussion. Authors broadly discussed issue of total suspended solids (TSS) (Lines 320-331) but their study did not deal with this aspect. So such part needn’t be so broad or the connection of TSS with this study must be explained.

Response: One of the factors that made this study unique was the use of GPS equipment to track equipment movement and determine their role in sediment path development. The graduate student that is the lead author on this manuscript was responsible for collecting GPS data from the equipment each day and performing the surveys for sediment path detection and associated activities. He was not responsible for measuring water quality, that was another student’s project. We have presented the information in the manuscript dozens of times at regional, national and international conferences. We have also discussed with forestry management groups and regulatory agencies. In nearly every occasion, we were asked about how the sediment paths affected water quality (what were the concentrations of TSS and turbidity?). Because it was commonly asked, we felt compelled to at least allude to studies that provided water quality information on the project (see references [30] and [31]). As such, we would like to keep the material in the paper so as readers can find the references if additional information is desired.

Chapter Discussion. Authors shows differences among skid trail morphology and number of sediments path (Table 2). Flat parts of trails cause the least number of sediment path. This statement is not discussed. Discussion is concentrated to width of streamside management zone or presence of logging debris but an easy way how to mitigate negative impact of timber transport is neglected. More careful design of forest skid trails whose slope gradient is less than 5% (flat) enables relatively harmless timber transport from clear-cut areas to log landings. Authors ignored such opportunity although they mentioned the fact that primary skid trails were installed along the contour (Page 4, Line 112).

Response: We understand the reviewers comment and have addressed with the following changes:  On lines 320 – 325 we added “low points along skid trails where water was accumulating from two directions exacerbated sediment path development. While, mitigation of sediment path development can be accomplished in skid trail construction by reducing the number of accumulation points, in highly dissected terrain there may be instances where this is unavoidable. In these cases, the proper placement of multiple water control structures to collect runoff before it accumulates would likely reduce sediment path development.”  And we deleted: “Water control structures at low points on a skid trail (not associated with ephemeral channels) that collect water from both sides of the structure are a specific issue that can potentially be alleviated with proper placement of multiple control structures at these microtopographic points.”

We also added the following to lines 386 and 387: “skid trail construction minimizes runoff accumulation points,”  

We would also like to note that there was an almost 240-m change in elevation from the edge of the SMZ to the landing deck. Although the contour skid trails were created to be as flat as possible, there were areas where skidding had to be done uphill to get to the landing. Lines 111 and 112 state: “All timber was skidded uphill to log decks located on the tops of ridges and hauling was conducted outside of the study watersheds.”

 Chapter Conclusion: Authors introduces that they found that harvesting and trail building on mesic slopes will lead to more sediment reaching streams (Page 14, Lines 364-365). However, this statement is not supported in their study. Only trail building lead to more sediment reaching streams. Harvesting means operations from tree felling to timber transport. But authors say that no sediment paths were found to originate from equipment disturbances off skid trails (Page 9, Lines 233-234).

Response: The reviewer’s comment is correct, we did not find a harvesting effect on sediment path development. The sentence was re-written as:  “We found that trail building on mesic slopes will lead to more sediment reaching streams, suggesting the value of increasing water control structure frequency on constructed skid trails, especially adjacent to SMZs on mesic slopes.”